# Burst-and-coast swimmers optimize gait by adapting unique intrinsic cycle

Gen Li [1], Intesaaf Ashraf[2], Bill François[2], Dmitry Kolomenskiy[3], Frédéric Lechenault[4],
Ramiro Godoy-Diana [2✉] & Benjamin Thiria [2✉]

This paper addresses the physical mechanism of intermittent swimming by considering the burst-and-coast regime of fish swimming at different speeds. The burst-and-coast regime consists of a cycle with two successive phases, i.e., a phase of active undulation powered by the fish muscles followed by a passive gliding phase. Observations of real fish whose swimming gait is forced in a water flume from low to high speed regimes are performed, using a full description of the fish kinematics and mechanics. We first show that fish modulate a unique intrinsic cycle to sustain the demanded speed by modifying the bursting to coasting ratio while maintaining the duration of the cycle nearly constant. Secondly, we show using numerical simulations that the chosen kinematics by correspond to optimized gaits for swimming speeds larger than 1 body length per second.

[1] Japan Agency for Marine-Earth Science and Technology (JAMSTEC), Yokohama, Japan. [2] Laboratoire de Physique et Mécanique des Milieux Hétérogènes (PMMH), CNRS UMR 7636, ESPCI Paris—PSL University, Sorbonne Université, Université de Paris, 75005 Paris, France. [3] Global Scientific Information and Computing Center, Tokyo Institute of Technology, Tokyo, Japan. [4] Laboratoire de Physique de l'École Normale Supérieure (LPENS), 75005 Paris, France. ✉email: ramiro@pmmh.espci.fr; bthiria@pmmh.espci.fr

ntermittent dynamics are frequently observed in fish loco-
motion. Known as burst and coast, the gait consists of a two-
step sequence, including an active phase during which fish
produce the propulsive force, followed by an inertial, passive
phase where they glide or coast without muscular action. This
behavior is observed either permanently as part of the strategy of
an animal to move and explore its environment, or during short
periods as part of high-speed swimming regimes. Burst and coast
have been addressed extensively by the biomechanics community
in the past decades[1–10], often associated with locomotion cost
optimization. Starting from the early studies of Weihs[1], these
works have essentially investigated the relationship between the
construction of the burst-and-coast cycle and the global swim-
ming efficiency, when compared to continuous undulatory
mechanisms.

Intermittent swimmers minimize the energetic cost of swim-
ming in the gliding phase, during which the fish body is passive
and straight, hence not producing mechanical effort and dis-
sipating less into the fluid. The energetically optimal working
point at a given speed is then obtained by tuning the typical times
spent in the burst and coast phases of the cycle, balancing the
advantage of the passive coasting phase and the energetic injec-
tion of the bursting phase to sustain the desired average speed.
Most studies have addressed this mechanism theoretically,
reducing the problem of burst-and-coast swimming to the opti-
mization of a mechanical system, decoupled from physiological
behavior[2,10]. Overall, there is a strong lack of experimental data
and parametric studies concerning intermittent swimmers,
making the description of speed modulation strategies for these
animals an open question.

The purpose of this paper is to examine burst-and-coast
swimming using an experimental work performed on live fish
that swim using the body and caudal fin (BCF) propulsion. A

typical burst-and-coast swimmer, the red-nose tetra fish *Hemi-
grammus bleheri*[11,12], is forced to swim in a flume at a given
velocity $U$, and video recordings are used to examine changes in
the gait as the imposed velocity $U$ changes. We show that, instead
of modulating the frequency and amplitude of the kinematics, the
fish rather adapt the burst-to-coast ratio keeping the time of a
typical burst-and-coast event within a narrow range between 0.2
and 0.4 s. The burst phase is a sequence of tail beats with nom-
inally constant frequency and amplitude. More importantly, we
demonstrate using a 3D numerical model based on the experi-
mentally measured swimming kinematics, that for a given
swimming speed, the burst-and-coast cycle chosen by the fish
corresponds to a gait minimizing the global cost of transport.

## Results

The experiments were conducted on four individuals of *Hemi-
grammus bleheri* fish in a water flume with flow velocity varying
from 0 to 3 body lengths per second (BL/s). Each fish is recorded
in runs of 10 s using fast camera imaging, and the body undu-
lation kinematics is subsequently characterized by extracting the
midline of the fish images for each video recording (see "Meth-
ods"). For each imposed swimming, the measurements were
repeated four times, giving a set of 16 different runs for compiling
one data point.

Figure 1 shows two typical cycles of swimming for a fish,
respectively, at imposed velocities of 1.15 BL/s and 1.9 BL/s. The
corresponding tail-tip kinematics are also plotted in Fig. 1d and e.
An additional case of lower swimming speed is shown in Fig. 1c,
to clearly define graphically the characteristic duration of the
burst and coast swimming cycle $T_{bout} = T_{burst} + T_{coast}$. A tail-beat
cycle refers to the fish completing two strokes, respectively, on
each side of the body in continuous swimming, while here in

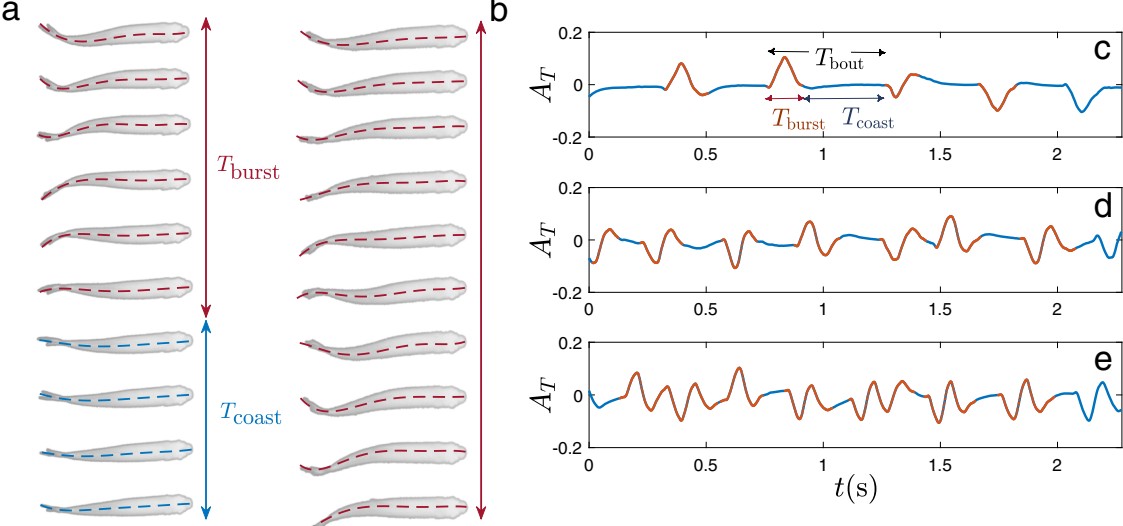

**Fig. 1 Burst-and-coast kinematics. a, b** Sequence of snapshots of typical swimming cycles for two different imposed swimming speeds, **a**: $U = 1.15$ BL/s
and **b** $U = 1.9$ BL/s . Time goes from top to bottom. The swimming cycle is composed of burst and coast phases. During the burst phase, the fish produces
propulsion by undulating its body, while during the *coast* phase the fish pauses and glides thanks to its own inertia. As can be seen, the main difference in
kinematics between the two gaits (**a**) and (**b**) is the time ratio between the propulsive action and the pause. The first part of the swimming cycles are
similar, with comparable beating frequency and amplitude of the body undulation but the fish swimming at the larger speed in (**b**) continues beating for the
full cycle while the other pauses before the next one. The midline extracted from the video post-processing is superposed to each frame. **c–e** Typical tail-tip
kinematics extracted from video analysis for three different imposed swimming velocities. **c** $U = 0.66$ BL/s, **d** $U = 1.15$ BL/s, and **e** $U = 1.9$ BL/s. As can be
observed, the intrinsic characteristics of the active burst cycle (in red color) share the same properties whatever the imposed gait. The frequency and
amplitude of the basic tail beat are similar and the imposed velocity is sustained just by increasing linearly the burst time with respect to the pause time.
Note that a tail-beat cycle consists of two strokes respectively towards each side of the body, thus in panel (**c**), a single stroke is counted as a half tail-
beat cycle.

intermittent swimming, we adopt this definition to measure the undulations inside the burst phase. The duration of bursts and coasts are measured from the tail-beating signals using the velocity and acceleration of the tail tip, which tend to zero during the coasting phase. We quantitatively defined the fish to be coasting when tail tip velocities and accelerations were lower than ~10% of their maximum values. A first observation to be made is that the tail-beat cycle of the fish is roughly the same regardless of the swimming speed, i.e., the tail-beat amplitude and frequency of the tailbeat remain visually similar. In order to sustain the increasing swimming velocities between frames c to d of Fig. 1, the strategy seems to be to increase the number of tail beats within the burst (note that as shown in Fig. 1c, a single stroke is counted as a half tail-beat cycle), while the coasting time is diminished. This observation is confirmed by the results obtained for all different individuals and swimming speeds. Figure 2 shows all the relevant quantities of the swimming kinematics, including the characteristic duration of the full burst and coast swimming cycle $T_{\text{bout}}$ in Fig. 2a; the duty cycle, i.e., the time ratio of the burst phase and the full burst-and-coast bout $DC = T_{\text{burst}}/T_{\text{bout}}$ (Fig. 2b); and the typical tail-beat frequency $F_i$ and scaled amplitude $\bar{A} = A/L$ in the burst phase—Fig. 2c and d, respectively.

In addition, the cost of transport CoT obtained from the numerical model is shown as a function of the swimming speed in Fig. 2e. The cost of transport is defined as

$$CoT = \frac{\bar{P}}{m\bar{U}},$$

i.e., as the power $\bar{P}$ normalized by the average swimming speed $\bar{U}$ and the mass of the fish (see ref. [13] and "Methods" for details). The shaded area in each subplot of Fig. 2 indicates the region of low swimming speed representing unusual slow gaits for this type of fish, which is discussed further in the text.

Figure 2 clearly shows that fish have three characteristic times on which they construct the burst-and-coast kinematics to attain the desired gait: $T_{\text{bout}}$, $T_{\text{burst}}$, and $T_i = 1/F_i$. The first characteristic time, $T_{\text{bout}}$, is constrained essentially to a range from 0.2 to 0.4 s for all fish—Fig. 2a. More formally, fitting a linear mixed-effects model that includes variation across individual fishes as a random effect suggests that $T_{\text{bout}}$ of a typical individual varies from 0.23 s at $U = 0$ to 0.35 s at $U = 3$ BL/s, i.e., by 52%. The fixed-effect slope $dT_{\text{bout}}/dU$ falls short of statistical insignificance ($P = 0.0364$). A similar fit in the range of low and medium speeds only ($U < 2$ BL/s) yields a smaller and statistically insignificant ($P = 0.3119$) estimate for the slope (see Supplementary Information, Part 1). Thus, it seems that the full burst-and-coast cycle depends weakly on of the swimming speed: fish do not modulate the time between two phases of action unless the imposed velocity is approaching the upper limit for sustained swimming, which is slightly above 3 BL/s in this experiment. Other works have already reported the regularity of the burst-and-coast swimming for different species and different stages of maturity (for instance, see refs. [14–18]), and this has been attributed to neural sensing mechanisms[17,19]. It is also worth noting that this time remains fairly constant across individuals. In contrast, Fig. 2b shows that the duty cycle DC (i.e., the fraction of the burst-and-coast bout during which the fish are actively

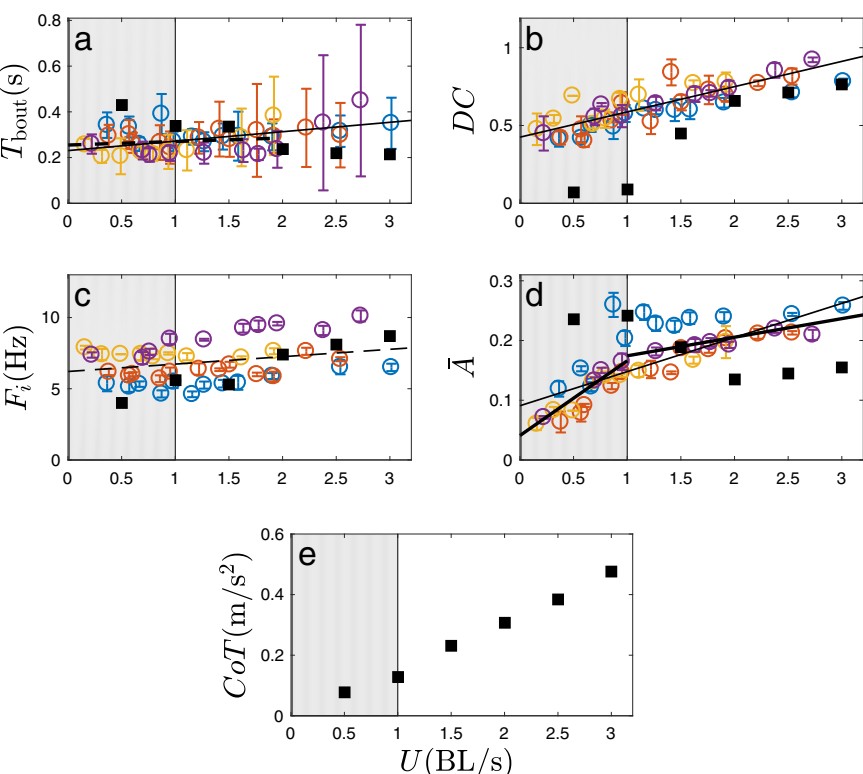

**Fig. 2 Burst-and-coast properties and cost of transport as a function of the imposed swimming speed $U$ in body length per second (BL/s). a** Bout duration $T_{\text{bout}}$. **b** Duty cycle DC = $T_{\text{burst}}/T_{\text{bout}}$. **c, d** Frequency and peak-to-peak amplitude of the tail beat during a bursting phase, respectively. **e** Cost and transport, CoT. Different marker colors correspond to different individuals. Error bars are standard deviations. The optimization results obtained from the burst-and-coast model based on 3D numerical simulations of an artificial swimmer are superimposed to the experimental data in filled black square symbols (■). Thick solid and dashed straight lines correspond to the linear mixed-effects model fits having, respectively, statistically significant and insignificant slopes. Panels (**a**) and (**d**) show global fits (thinner lines) and local piece-wise fits (thicker lines). See Supplementary Information (Part 1) for the source experimental data points for each fish.

producing thrust) increases linearly with the demanded swimming speed (linear mixed-effect model fit produces a significant fixed-effect slope, $P = 8.524 \times 10^{-5}$; simple linear regression yields $P = 8.754 \times 10^{-13}$ and adjusted $R^2 = 0.7095$). The other characteristic time $T_i = 1/F_i$ is the inverse of the internal frequency $F_i$ shown in Fig. 2c. The absolute value of $F_i$ seems related to each specific fish, but it is statistically constant for all velocities tested in the experiment ($P = 0.06847$ for the fixed-effect slope of a global mixed-effects model fit), meaning that fish do not modulate this internal tail-beat frequency as a gait adapting strategy. Concerning the amplitude of the burst cycle, Fig. 2d shows that the magnitude of the tail beat increases in the range of slow velocity (0 BL/s $< U < 1$ BL/s piece-wise fit, $d\overline{A}/dU = 0.12519$ (BL/s)$^{-1}$, $P = 0.0001866$) to saturate afterward at higher speeds ($d\overline{A}/dU = 0.031129$ (BL/s)$^{-1}$, $P = 0.007688$). This can be readily understood recalling that the burst is short at low velocities so that there is no time to accommodate more than one half of a period of the tail-beat oscillation—see the burst profiles in Fig. 1c. The amplitude change between low speeds and medium-high speeds is further supported by pairwise $t$ tests (see Supplementary Information, Part 1).

## Discussion

The reading of Fig. 2 tells that intermittent swimmers repeat an intrinsic basic movement to sustain the desired swimming speed. This movement consists of an active undulation of almost constant frequency and almost constant tail-beat amplitude (except in the low-speed range, shaded in gray in the panels of Fig. 2), repeated as long as it is needed. Thus, a fish willing to swim faster will increase its bursting time. Of course, because each burst-and-coast swimming sequence is performed over an almost constant time $T_{bout}$, fish spending more time in the burst phase necessarily also shorten the coast duration, which sets an upper limit to the swimming speed that can be achieved. It is interesting to note that the swimming behavior described here differs from the idea that fish modulate their body wave kinematic parameters to change speed, in contrast with what has been observed for larger fish using continuous swimming—for instance, see refs. [20,21]. To our knowledge, such a mechanism has not been reported in the literature, especially concerning small-sized fish of a few centimeters as the tetra fish of the present experiments.

In order to understand the dynamics underlying the experimental observations, we studied the swimming optimization problem of a simulated burst-and-coast swimmer. The fish is modeled using the realistic body geometry of *Hemigrammus bleheri* extracted from the experiment (see Supplementary Information, Part 2). The burst-and coast cycle is built, following the observations, by concatenating an active phase and a passive phase. The flow field around the fish during each phase is simulated using computational fluid dynamics (CFD)—see "Methods". Through exploring the parameter space, for each swimming velocity, the set of parameters (DC, $F_i$, $\overline{A}$, $T_{bout}$) that minimizes the cost of transport (CoT) is selected. The results of the optimization procedure are superimposed to the experimental data in Fig. 2 (black squares).

For moderate-to-high speeds (in the range 1 BL/s $< U < 3$ BL/s), the parameters that minimize the energy cost of swimming match closely the experimental data, especially in terms of $T_{bout}$, DC, and $F_i$, the predicted points overlap the observation data. As to $\overline{A}$, the predicted results are lower than the observations, probably because: (1) the factual fish deformation cannot be precisely reproduced by the sinusoidal function used for the numerical model; (2) when modeling the red-nose tetra fish, we simplified its forked tail by a triangular one, hence the model fish tail, with a slightly larger area, may generate relatively stronger lateral recoil,

neutralizing the effective amplitude. Nevertheless, the prediction on $\overline{A}$ still suggests that fish does not rely on amplification of the tail-beat amplitude to propel faster, which is similar to the trend observed in the experiment.

This is a remarkable observation, as it shows that fish in the range of cruise swimming speeds constantly optimize their CoT. Optimality is not straightforward in the multidimensional space navigated by living organisms, where locomotion is just one element of their everyday trade-offs. More specifically, the observation and its agreement with the simulation are exciting for a double reason. In the first place, unlike in continuous swimming where fish basically deal with a two-dimensional parameter space consisting of tail-beat frequency and amplitude, in burst-and-coast swimming, fish have to deal at least with a four-dimensional parameter space (shown in Fig. 2: $T_{bout}$, DC, $F_i$, and $\overline{A}$) at an arbitrary speed. The optimization of burst-and-coast swimming is thus extremely complex, especially considering that the CoT can hardly be sensed directly by the fish during swimming. Second, fish have to deal with many other constraints that might not be, a priori, necessarily compatible with optimizing swimming energy. For instance, the intermittence of burst-and-coast swimming has also been invoked for a sensing reason[17]. Before the present work, we did not know whether fish aim to optimize the CoT during burst-and-coast swimming, or whether fish can successfully optimize CoT in such a complex landscape of control parameters and indirect feedback. It turns out that in the case of this work, the intermittent swimming kinematics is, in a certain range of swimming regimes, exactly what optimizes swimming gaits. It is also surprising that fish can handle the optimization of CoT in burst-and-coast swimming relatively easily—such optimization mainly consists in maintaining the tail-beat frequency and amplitude constant and modulating the time of bursting.

The remarkable agreement between the optimization calculation and experimental observations leads us to two important conclusions for burst and coast swimmers. First, fish essentially do not modulate tail-beat frequency as observed for continuous swimming[20,21] but adapt a unique cycle to sustain the imposed speed. Second, the frequency, amplitude of the tail beat, and the burst phase duration (the duty cycle) are optimal parameters with respect to the cost of transport CoT at typical cruise speeds. It is also noteworthy that the results of the simulation are not exclusively associated to the species *Hemigrammus bleheri*. Excepted the details of the body shape that were extracted from the experiments, the construction of the intermittent simulated kinematics (see "Methods") uses a generic body deformation that can describe other burst-and-coast swimmers. The results presented in this paper bring a general description of intermittent fish locomotion, based on experimental observations: because of the intermittency constraint—the bout time, most likely fixed because of physiological reasons, these fish have developed specific swimming sequences minimizing their cost of transport that are different from those observed for continuous swimmers, and such specific swimming sequences do not require the fish to handle all optimal parameters in a complex pattern. Future works should multiply experimental observations and produce a larger inventory of intermittent swimmers to determine if the burst-and-coast mechanism described here holds for other fish species. It has to be noted that the CoT as defined in this study only takes into account the mechanical cost of the swimmers, thus future explorations on the consequence of considering the additional "metabolic" cost may bring us a more comprehensive understanding of the swimming cost and optimization in burst-and-coast swimmers.

However, the predictions of the optimization procedure fail to reproduce the observations in the low-speed range (0 BL/s $< U <$

1 BL/s). The optimization predicts larger $T_{\text{bout}}$ and $\bar{A}$, as well as a smaller DC. Such divergence may be caused by burst-coast transition factors beyond our numerical model. To our knowledge, at least three transition factors may produce errors. First, at the same speed, the drag on an undulating fish body is much higher than that on a gliding fish body. When fish stops undulation and starts gliding, its boundary layer transits from an undulating pattern to a gliding pattern, while fish stops gliding and start undulating, the fish boundary layer transits reversely. Such a drag transition is not modeled in our prediction. Second, when a fish starts the burst phase, it needs to initiate both its body deformation and acceleration of its surrounding water, hence the initial tail-beat may not generate thrust as effectively as succeeding tail-beats. Third, in reality, there are fluctuations in the swimming speed of fish. Fish may utilize the fluctuation of speed by ending its burst process at peak speed, in order to initialize a coast phase with additional momentum and further reduce CoT. These transition factors between the burst- and the coast phases may become relatively stronger in the low-speed range, as the fish performs only a single or two strokes during the burst phase. As the result, our numerical model becomes much less effective in low speed. On the other hand, the divergence at low swimming speeds between predictions and experiments may still have other possible explanations—such as the physiological constraints of muscle efficiency or the sensorimotor capacity necessary for maintaining the body orientation, but it may also be explained considering that CoT minimization might not be needed at such low swimming velocities due to the absolute low energy consumption.

## Methods

**Animals and housing**. Red nose tetra fish *Hemigrammus bleheri* of body length in the range ~3.5–4-cm long and height ~0.5–0.6 cm, were procured from a local aquarium supplier (anthias.fr, France). The fish were reared in a 60-liters aquarium tank with water at a temperature between 26 and 27 °C and they were fed five to six times a week with commercial flake food. Results from experiments with four individuals are analyzed here, for which a full set of different swimming speeds were recorded. The experiments performed in this study were conducted under the authorization of the Buffon Ethical Committee (registered to the French National Ethical Committee for Animal Experiments no. 40).

**Swimming flume**. A shallow-water tunnel with a test section of 2.2-cm depth and a swimming area of $20 \times 15$ cm was used for the experiments—see also refs. [11,12], where the same setup has been used to study collective swimming dynamics. The flow rate $Q$ can be varied from 4 to 22 liters per minute, resulting in an average velocity $U = Q/S$, where $S$ is the cross-section, in the range between 2.7 cm s$^{-1}$ and 15 cm s$^{-1}$ corresponding to Reynolds numbers $Re = UL/v$, $950 < Re < 6000$. The mean turbulence intensity in the channel is between 3 and 5% (characterized using PIV in the previous work[12]) and is independent of the flow rate. The velocity profile in the mid-section of the channel is rather flat and also remains unchanged for the different flow rates used, the wall effect region being limited to a distance smaller than 3 mm.

**Experimental procedure**. Before each measurement, the fish group was transferred to the swimming tunnel with the fluid at rest and left for around 1 h in order to acclimatize to the conditions of the experiments. The swimming runs were carried out for 10 s on each individual, increasing gradually the imposed speed from 0.5 BL/s to 3 BL/s. The procedure was then repeated several times, with a typical 30-min resting pause between measurements. The complete set of experiments consisted of three to five runs per individual, at ten different velocities, on a group of four individuals, corresponding to 150 measurement points. Video sequences at 400 frames per second were recorded using a Phantom Miro M-120 with 2 M pixel ($1920 \times 1200$) resolution and 12-bit pixel depth. Video post-processing was performed using an in-house Matlab code to extract the fish midline kinematics.

**Statistics and reproducibility**. The tail-beating kinematics was extracted for each fish in a group. The average and standard deviation were computed for each fish and each velocity. All points presented in Fig. 2 are thus averaged quantities with several experiments per point (individual data points for each of the individual fish can be found in Supplementary Section 1.1). For instance, the data points for the

frequency are given by $F_i = \frac{1}{N} \sum_0^N \langle f \rangle$ over the number of individuals and the different runs, where the brackets denote a time average and $N$ is the number of individuals within the school. Thus, we obtain a single value of the frequency for each fish and swimming speed.

Further statistical analyses have been performed to detect any significant variation of $T_{\text{bout}}$, $F_i$, $\bar{A}$, and DC with the swimming speed. The dataset consisted of the statistical averages of those four dependent variables, one value for each individual for each speed. Linear regression, linear mixed-model analyses, ANOVA and pairwise $t$ tests have been applied. All statistical calculations were performed in R (version 3.4.4).

**Burst-and-coast numerical model**. We developed a numerical model that can generate an arbitrary burst-and-coast swimming gait in a four-dimensional parameter space. The parameters are (1) the frequency of the burst phase $f_b$, (2) the amplitude of the burst phase $A_b$, (3), the upper speed bound $U_U$ (the speed at which the fish stops bursting and starts coasting), and (4) lower speed bound $U_L$ (the speed at which fish stops coasting and starts bursting, $U_L < U_U$). Then, we search across this parameter space for an optimal burst-and-coast swimming gait that guarantees sustained swimming with some specified speed $\bar{U}$ at the lowest cost of transport CoT. The numerical solutions of this constrained optimization problem involve a coarse discretization of the parameter space, a composition of a database of different gaits with those few discrete values of the frequency and amplitude, and a subsequent interpolation using that database.

The data for the burst phase are obtained by means of CFD simulations using a well-validated three-dimensional solver based on the overset-grid finite-volume method[22,23]; for more information including the numerical validation, see Supplementary Information, Part 2. We simulated "full burst processes" of a self-propelled fish in continuous swimming with some constant frequency and amplitude. The body length of the model fish is 2 cm, and its deformation is driven by a sinusoidal function. The fish accelerated from rest in quiescent water until it nominally reached its maximum speed—Fig. 3a. Since such a "full burst process" is fully determined by the tail-beat frequency and amplitude of the fish, we simulated 25 cases with five different frequencies (2, 6, 10, 14, and 18 Hz) and five different tail-beat amplitudes (~0.02, 0.07, 0.13, 0.19, and 0.26 $L$). The range of the Reynolds number in this study is below 6000, turbulence models are not used, and the grid resolution at $Re = 6000$ has been justified in a previous study[24]. The time sequence data of speed, power, and CoT from all full burst process cases were low-pass filtered to remove the periodic fluctuation caused by the tail beat. Using these 25 cases as interpolation nodes, one can quantify any arbitrary full burst process with some specified tail-beat amplitude and frequency—see Fig. 3b.

The coast phase motion and energetics data were obtained using the same CFD solver mentioned above, letting the model fish stop undulating after reaching the speed of 13 BL/s (the highest speed reached across all simulated cases corresponding to $f_b = 18$ Hz and $A_b = 0.26$ $L$). During this coast phase, the body was held straight and the fish decelerated until the velocity dropped to almost zero. Note that the mechanical power consumption in the coast phase is zero.

Thus, a burst-and-coast process is defined when an upper speed bound $U_U$ and a lower speed bound $U_L$ are specified. The full-burst and the full-coast data sequences are trimmed according to the values of $U_U$ and $U_L$, respectively. The full swimming cycle was obtained by concatenating the trimmed burst and coast time sequences considering that the transition between the burst and the coast phases is instantaneous. The procedure is then duplicated to produce a sawtooth-wave time profile of the velocity—see Fig. 3c.

For a given set of the four parameters ($f_b$, $A_b$, $U_U$, and $U_L$), as long as $U_U$ and $U_L$ are within the speed range of the "full burst process", we obtain a unique burst-and-coast swimming gait. The average speed of the generated burst-and-coast swimming gait is defined as $\bar{U}$ and the average power as $\bar{P}$. We programmed a MATLAB code to scan the four parameter dimensions in order to find an optimal burst-and-coast swimming gait that would meet the required speeds with the lowest cost of transport ($f_b$, scan resolution 1 Hz, range 2–18 Hz; $A_b$ scan resolution ~0.0015 $L$, range ~0.02–0.26 $L$; the scan resolution in $U_U$ and $U_L$ is less than $10^{-7} L$/s).

For further details of the numerical model, see Supplementary Information, Part 2.

**Definition of power**. In this paper, power refers to "mechanical power", defined as the sum of the hydrodynamic and body inertial powers:

$$P = P_{\text{hydro}} + P_{\text{body}}.$$

Hydrodynamic power is calculated as the sum of the hydrodynamic work on the body surface, such that:

$$P_{\text{hydro}} = \iint_{\text{surface}} (\mathbf{f} \cdot \mathbf{U}) ds,$$

where $P_{\text{hydro}}$ is the hydrodynamic power; $ds$ denotes surface element, $\mathbf{f}$ is the hydrodynamic stress vector acting on the surface element; $\mathbf{U}$ is the velocity vector on this surface element. Body inertial power is computed as the sum of the kinetic

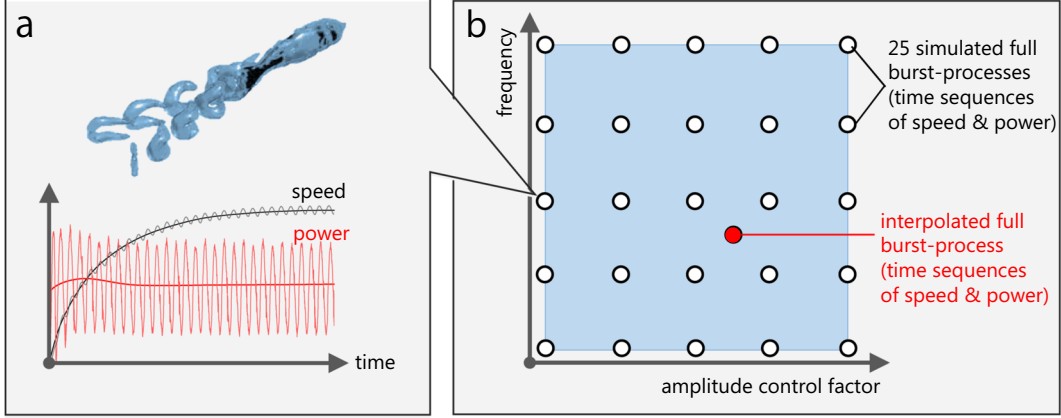

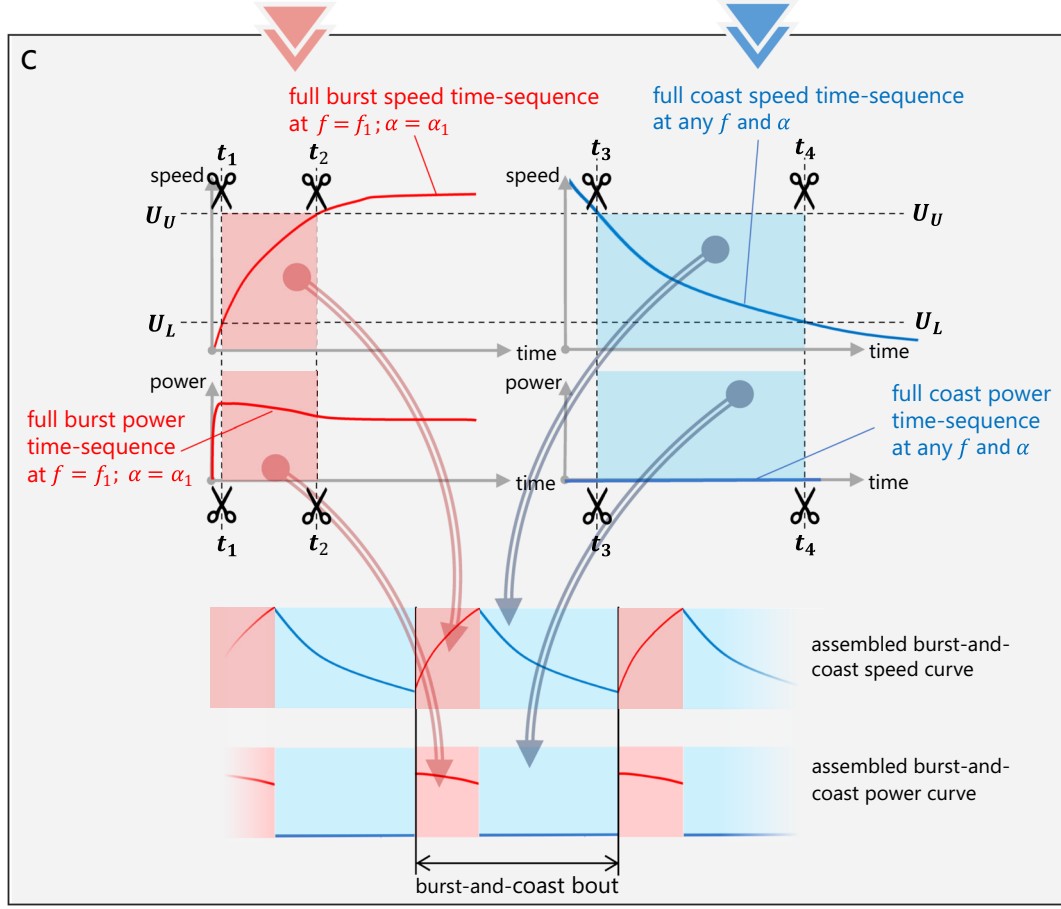

**Fig. 3 Graphical representation of the numerical burst-and-coast modeling. a** Simulation of a full burst process based on a three-dimensional self-propelled fish swimming model. In each simulation, the fish accelerates from a static condition until a stable speed is reached. The time sequences of speed and power are recorded. **b** Based on 25 sample simulations, for an arbitrary combination of tail-beat frequency and amplitude, the corresponding full burst process (time sequences of speed and power) is obtained by interpolation. **c** Full-burst process and full-coast process are trimmed and assembled into the burst-and-coast gait.

energy change rate of all body elements (inside the body), such that:

$$P_{\text{body}} = \iiint_{\text{body}} (\rho\mathbf{a} \cdot \mathbf{U})\mathrm{d}v,$$

where $P_{\text{body}}$ is the body inertial power; d$v$ denotes body volume element, $\mathbf{a}$ and $\mathbf{U}$ are the acceleration and velocity vectors of each body volume element; $\rho$ is the local density. Note that based on the previous equations, during the coast phase, the mechanical power is zero.

**Reporting summary.** Further information on research design is available in the Nature Research Reporting Summary linked to this article.

## Data availability

The datasets for the statistical analyses and the source data for Fig. 2 are available in the following Open Science Framework repository: https://osf.io/28bma/ (https://doi.org/10.17605/OSF.IO/28BMA). Source video sequences are available from the corresponding authors upon request.

## Code availability

The burst-and-coast gait assembly and the optimization algorithm are available in the following Open Science Framework repository: https://osf.io/28bma/ (https://doi.org/10.17605/OSF.IO/28BMA). The CFD solver of a three-dimensional fish swimming model has been described in detail elsewhere[22–24].

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

## Acknowledgements

We thank Prof. Hao Liu for his valuable contribution on the CFD model; the personnel from the PMMH Laboratory workshop, in particular Xavier Benoit-Gonin, for their technical help in the construction of the swimming channel; and José Halloy for his help with the fish handling protocol. G.L. was funded by the Japan Society for the Promotion of Science, Grant-in-Aid for Scientific Research (KAKENHI), 20K14978 and 17K17641.

## Author contributions

G.L. developed the burst-and-coast numerical model and the optimization code. I.A. and B.F. performed the experiments. D.K. performed statistical analyses. F.L., I.A., R.G.D., and B.T. post-processed and analyzed the experimental data. I.A., R.G.D., and B.T. designed the experiment. G.L., D.K., R.G.D., and B.T. defined the burst-and-coast model and wrote the paper. All authors approved the paper.

## Competing interests

The authors declare no competing interests.
