## [Peer Review File · Communications Biology]

Reviewers' comments:

Reviewer #1 (Remarks to the Author):

In this paper, the authors' present results recorded from live fish swimming and numerical models which show the studied intermittent swimmer not changing its body wave kinematics to change speed. The fish increase their average speed by increasing the burst time relative to a resting coast period. The results indicate that the fish seem to optimize their gaits to lower the cost of transport at moderate to high speeds. The paper does a good job of backing its experimental findings with a parametric numerical study. Additionally, it's easy to read and the ideas seem to flow coherently throughout the paper.

The main result seems to be the statement that at high speeds, burst and coast swimmers minimize the energy cost by selecting the appropriate parameters. The authors state, "This is a remarkable observation, as it shows that fish in the range of cruise swimming speeds constantly optimize their CoT." I'm not sure I agree with the strength of this statement. Assuming the species evolved to swim with some sort of efficiency in this range, it's hardly surprising. It sort of strikes me as an "it works because it works" type of result. Especially given that the authors state that the optimal CoT involves swimming with a consistent tail beat frequency and amplitude. It's just as likely that biomechanics determine these parameters, and at some range of swimming speeds, they become optimal.

Overall, the study is well done and interesting, however, for just a single main result, more work may need to be done to convince me of its importance.

Minor concerns

Section II notes that the swimming speeds for figure 1 are 1.15 and 1.65 BL/s, however, the caption of Fig. 1 states 1.15 and 1.9 BL/s. Please correct whichever is incorrect.

Introduction Page 1 paragraph 1 line 8: Misspelled 'environment'

Introduction Page 1 column 2: 'We show that, instead of modulating the frequencyand amplitude'

Define Tbout when you first mention it.

Figure 2: When talking about figure 2 in results, it would be easier to talk about them in the way they are listed. So maybe move the duty cycle to figure D or talk about it before.

Methods Section C: 'Bodylength of modle fish is 2cm' (typo in "model")

There are spaces before ":" throughout the paper (i.e. "swimming kinematics : the"). Remove the space. Also, there are lots of colons in the paper. Maybe the authors should consider a more diverse rhetorical/grammatical repertoire.

Discussion, paragraph 3: In the final sentence of the paragraph, remove the comma after the m-dash. This happens again in the following paragraph. Perhaps this is an unconventional style of the journal, which will, of course, refer to its own style guide. I'm just noting it, as I have not encountered it before.

Reviewer #2 (Remarks to the Author):

This paper focuses on intermittent swimming – a mode of fish locomotion where the fish actively beats its tail in a "burst" before passively gliding in a "coast." In contrast to continuous, steady

swimming gaits, which have received extensive treatment laying a solid framework for more advanced study, intermittent gaits like burst-and-coast motion have not been studied so thoroughly. Real fishes tend to swim in complex, unsteady gaits like burst-and-coast motion during key behaviors like foraging. So, this study represents a crucial first step that will enable a more complete understanding of fish swimming, both from a biological perspective focusing on the influence of swimming ability on body plan evolution and ecological roles, and from an engineering perspective applying principles of aquatic animal motion to underwater vehicle design.

Notably, the authors have used both experimental and computational investigations in a highly complementary manner. These approaches are likely to appeal to a broad audience including both biologists and engineers. Particularly, the authors use simulations to quantify the swimming performance achieved through broad range of kinematic options including those outside a fish's natural behavioral repertoire. The authors use this "performance space" to try to explain, mechanistically, why real fish swim the way they do – that is, the study at its core is based on biological questions.

I think this manuscript represents an excellent bit of science, and I do have two major comments, and several minor suggestions, that would help clarify and strengthen the authors' arguments.

First, there are several places throughout the manuscript where the text is difficult to understand – particularly in the methods. The computational work has been described quite thoroughly in the supplements, and I commend the authors on their inclusion of Fig. 3 in the main text, which greatly clarifies how the burst-and-coast gait was constructed from the larger simulations. The treatment of the experimental work is less thorough, and at times, I had difficulty understanding what exactly the authors did. I believe the methods are sound – but some revisions for clarity would greatly facilitate reader understanding. Specifics can be found in the line-by-line section below.

Second, at times, the authors comment that certain values are "no different" and imply that this is statistically so, but no statistical analyses appear to have been performed or reported. Formal statistical analyses corroborating these assertions would greatly strengthen the authors' conclusions. Again, specific suggestions can be found in the line-by-line section below. I would encourage the authors to consult Harrison et al. (2018) and Zuur et al. (2009) (or similar) for background information on the specific statistical analyses suggested and their interpretation.

Line-by-line comments:

1. Introduction, end of second paragraph ("Overall, there is...open question") – "gait adaption" should be "gait adaptation." The original is not wrong, but "adaptation" is the more accepted spelling.
2. Introduction, end of second paragraph ("Overall, there is...open question") – unclear what is meant by "gait adaption in real animals is an open question." A sentence or two expanding on what the authors mean would help reader understanding. Is the point of this paragraph that burst-and-coast swimming is often treated as an optimization problem in computational settings, but it is unclear if real animals use optimal cadences of movement?
3. Figure 1, panel B – The fish silhouettes in panel B are somewhat confusing. Specifically, the fish's tail jumps from an extreme lateral excursion toward the bottom of the page in the 5th panel to nearly the opposite lateral extreme in the 6th panel. If this reflects real motion at some specific time interval, then a note that the tailbeat motion is unsteady, so you cannot expect the same amount of movement between panels, would greatly aid reader understanding.
4. Figure 1, panel E – While burst/coast bouts are pretty clear in C and D, it is unclear how burst-and-coast patterns are defined in this panel. For example, can we really tell that the tiny blue tick

at $t = 2.5$ is a "coast"? Is it typical that the number of half-tailbeats in a bursting period are this irregular?

5. Figure 1 caption and Results paragraph 2 – how is a tailbeat being defined? A tailbeat cycle consists of a complete sine wave of amplitude change, based on the motion of the tail tip. From Figure 1C, it looks like the authors may be using "tailbeat" to refer to half a tailbeat cycle. An explicit definition would greatly clarify the authors' meaning.

6. Results, paragraph 3 (starts with "In addition, the cost of transport...") It would be helpful to have some explanation of how power is measured. Is this based on a calculation of energy consumption in the simulation? Based on forces and velocities from the simulation? (Etc.)

7. Results, paragraph 4 (starts with "Fig. 2 clearly shows...") The time range for T_{bout} is small in absolute terms (0.2s), but a change from 0.2s to 0.4s represents a doubling in time. How does this play out over swimming speed? If the smaller values all are associated with the slowest swimming speed, then it would mean that T_{bout} is not constant. This is a place where a formal statistical analysis would support the authors' claims. For instance, if there is no change in T_{bout} with swimming speed, then a linear regression should have a non-significant slope (i.e., a hypothesis test indicates that we cannot conclude there is a relationship between swimming speed and T_{bout}). A non-significant regression slope in a linear mixed effects model, which would account for variation across individual fishes by including individual as a random effect, would provide the necessary support.

8. Results, paragraph 4 (starts with "Fig. 2 clearly shows...") As with T_{bout} in comment #6, the authors claim that frequency is "statistically constant," but no detail on statistical analyses has been provided. A similar linear mixed effects modeling approach is suggested.

9. Results, paragraph 4 (starts with "Fig. 2 clearly shows...") Again, no statistical support has been provided for claims about DC or amplitude. Linear mixed effect models again would assist with the DC analysis. For amplitude, the authors may need to consult with a statistician to discuss the best approach to statistically support the change in slope indicated.

10. Figure 2 – A legend indicating the meaning of the colors and shapes would greatly facilitate reader understanding. And, what does the shading between speeds 0-1 BL/s indicate? Do the error bars represent standard error or standard deviation?

11. Figure 2 – The amplitude of the simulation does not match real fish tailbeat amplitude particularly well (overestimating amplitude between 0-1 BL/s, then decreasing and underestimating amplitude during higher swimming speeds). To be clear, I think the deviation between simulation and reality is within expectations, I do not believe this has been discussed in the manuscript – where does this difference come from? What impact may it have on the cost of transport analysis?

12. Discussion, paragraph 3 ("...it may also be explained considering that CoT minimization might not be needed at such low swimming velocities...") I would further encourage the authors to consider how often fish swim in this mode at low speeds – if it is uncommon, then perhaps there has not been selection for minimizing energy costs. Is this the case?

13. Methods, section A – It would be helpful to have a sense of the Reynold's number for the live fishes here. The authors later report the Re range for the simulation, but a value range for the experimental data would help readers understand how well the simulation matches the experimental conditions.

14. Methods, section B – The flume here is extremely shallow – only 3-4 times deeper than the fish's body depths. The potential for wall and surface effects here concerns me. The authors do

mention small wall effects, but it is unclear whether this is relative to the sides or bottom of the tank. And, there is no mention of whether surface effects are important. Likewise – where were the fish swimming in the water column? With a tank this shallow, the fish would need to be very centered (vertically) to avoid wall and surface effects.

15. Methods, section C – Unclear what is meant by “groups.” Were multiple fishes studied at once? In that case, did the fishes school? Which fish in the school did the data come from? Because the wake of fish at the front of a school influences flow felt by fish in the back, and because we know that fish kinematics change when entrained on vortices like those in fish wakes (see Liao 2007, Fig 1), average kinematics across a school would not reflect what an individual fish does during burst-and-coast swimming.

16. Methods, section E, paragraph 2 – What data were filtered? CoT?

17. Methods, section E, paragraph 4 – An assumption baked into this burst-and-coast assembly process is that the transition between phases is instantaneous. But in real fishes, there would be some kind of transitional flow/wake as the fish switches between bursting and coasting. My understanding is that these would be very difficult – if not completely impractical – to simulate, so I do believe that the authors have used an appropriate approximation with this assembly approach. Even so, it would be helpful to see some comments explaining the limits of the model and how these differences may influence the performance of real fishes, especially since the key conclusion drawn is that real fishes are using optimal kinematics.

18. General – it would be helpful for readers of more biological backgrounds if the authors made a distinction between mechanical costs of transport (those studied here) and metabolic costs of transport. While a fish’s gait may be optimal in the mechanical sense, that does not necessarily mean that the gait is simultaneously providing the lowest metabolic cost. I do not think this distinction lessens the importance of this work – rather, it provides more context and sets up more ways this work can fit into future investigations.

References:

Harrison XA, Donaldson L, Correa-Cano ME, Evans J, Fisher DN, Goodwin CED, Robinson BS, Hodgson DJ, and Inger R. 2018. A brief introduction to mixed effects modelling and multi-model inference in ecology. *PeerJ* 6: e4794.

Liao J. A review of fish swimming mechanics and behavior in altered flows. *Phil Trans R Soc B* 362: 1973-1993.

Zuur AF, Ieno EN, Walker N, Saveliev AA, and Smith GM. 2009. *Mixed effects models and extensions in ecology with R*. New York: Springer.

In this paper, the authors' present results recorded from live fish swimming and numerical models which show the studied intermittent swimmer not changing its body wave kinematics to change speed. The fish increase their average speed by increasing the burst time relative to a resting coast period. The results indicate that the fish seem to optimize their gaits to lower the cost of transport at moderate to high speeds. The paper does a good job of backing its experimental findings with a parametric numerical study. Additionally, it's easy to read and the ideas seem to flow coherently throughout the paper.

The main result seems to be the statement that at high speeds, burst and coast swimmers minimize the energy cost by selecting the appropriate parameters. The authors state, "This is a remarkable observation, as it shows that fish in the range of cruise swimming speeds constantly optimize their CoT." I'm not sure I agree with the strength of this statement. Assuming the species evolved to swim with some sort of efficiency in this range, it's hardly surprising. It sort of strikes me as an "it works because it works" type of result. Especially given that the authors state that the optimal CoT involves swimming with a consistent tail beat frequency and amplitude. It's just as likely that biomechanics determine these parameters, and at some range of swimming speeds, they become optimal.

Overall, the study is well done and interesting, however, for just a single main result, more work may need to be done to convince me of its importance.

Minor concerns

Section II notes that the swimming speeds for figure 1 are 1.15 and 1.65 BL/s, however, the caption of Fig. 1 states 1.15 and 1.9 BL/s. Please correct whichever is incorrect.

Introduction Page 1 paragraph 1 line 8: Misspelled 'environment'

Introduction Page 1 column 2: 'We show that, instead of modulating the frequencyand amplitude'

Define T_{bout} when you first mention it.

Figure 2: When talking about figure 2 in results, it would be easier to talk about them in the way they are listed. So maybe move the duty cycle to figure D or talk about it before.

Methods Section C: 'Bodylength of modle fish is 2cm' (typo in "model")

There are spaces before ":" throughout the paper (i.e. "swimming kinematics : the"). Remove the space. Also, there are lots of colons in the paper. Maybe the authors should consider a more diverse rhetorical/grammatical repertoire.

Discussion, paragraph 3: In the final sentence of the paragraph, remove the comma after the m-dash. This happens again in the following paragraph. Perhaps this is an unconventional style of the journal, which will, of course, refer to its own style guide. I'm just noting it, as I have not encountered it before.

Comments to the authors

This paper focuses on intermittent swimming – a mode of fish locomotion where the fish actively beats its tail in a “burst” before passively gliding in a “coast.” In contrast to continuous, steady swimming gaits, which have received extensive treatment laying a solid framework for more advanced study, intermittent gaits like burst-and-coast motion have not been studied so thoroughly. Real fishes tend to swim in complex, unsteady gaits like burst-and-coast motion during key behaviors like foraging. So, this study represents a crucial first step that will enable a more complete understanding of fish swimming, both from a biological perspective focusing on the influence of swimming ability on body plan evolution and ecological roles, and from an engineering perspective applying principles of aquatic animal motion to underwater vehicle design.

Notably, the authors have used both experimental and computational investigations in a highly complementary manner. These approaches are likely to appeal to a broad audience including both biologists and engineers. Particularly, the authors use simulations to quantify the swimming performance achieved through broad range of kinematic options including those outside a fish’s natural behavioral repertoire. The authors use this “performance space” to try to explain, mechanistically, why real fish swim the way they do – that is, the study at its core is based on biological questions.

I think this manuscript represents an excellent bit of science, and I do have two major comments, and several minor suggestions, that would help clarify and strengthen the authors’ arguments.

First, there are several places throughout the manuscript where the text is difficult to understand – particularly in the methods. The computational work has been described quite thoroughly in the supplements, and I commend the authors on their inclusion of Fig. 3 in the main text, which greatly clarifies how the burst-and-coast gait was constructed from the larger simulations. The treatment of the experimental work is less thorough, and at times, I had difficulty understanding what exactly the authors did. I believe the methods are sound – but some revisions for clarity would greatly facilitate reader understanding. Specifics can be found in the line-by-line section below.

Second, at times, the authors comment that certain values are “no different” and imply that this is statistically so, but no statistical analyses appear to have been performed or reported. Formal statistical analyses corroborating these assertions would greatly strengthen the authors’ conclusions. Again, specific suggestions can be found in the line-by-line section below. I would encourage the authors to consult Harrison et al. (2018) and Zuur et al. (2009) (or similar) for background information on the specific statistical analyses suggested and their interpretation.

Line-by-line comments:

1. Introduction, end of second paragraph (“Overall, there is...open question”) – “gait adaption” should be “gait adaptation.” The original is not wrong, but “adaptation” is the more accepted spelling.
2. Introduction, end of second paragraph (“Overall, there is...open question”) – unclear what is meant by “gait adaption in real animals is an open question.” A sentence or two expanding on what the authors mean would help reader understanding. Is the point of this paragraph that burst-and-coast

swimming is often treated as an optimization problem in computational settings, but it is unclear if real animals use optimal cadences of movement?

3. Figure 1, panel B – The fish silhouettes in panel B are somewhat confusing. Specifically, the fish's tail jumps from an extreme lateral excursion toward the bottom of the page in the 5th panel to nearly the opposite lateral extreme in the 6th panel. If this reflects real motion at some specific time interval, then a note that the tailbeat motion is unsteady, so you cannot expect the same amount of movement between panels, would greatly aid reader understanding.
4. Figure 1, panel E – While burst/coast bouts are pretty clear in C and D, it is unclear how burst-and-coast patterns are defined in this panel. For example, can we really tell that the tiny blue tick at $t = 2.5$ is a “coast”? Is it typical that the number of half-tailbeats in a bursting period are this irregular?
5. Figure 1 caption and Results paragraph 2 – how is a tailbeat being defined? A tailbeat cycle consists of a complete sine wave of amplitude change, based on the motion of the tail tip. From Figure 1C, it looks like the authors may be using “tailbeat” to refer to half a tailbeat cycle. An explicit definition would greatly clarify the authors' meaning.
6. Results, paragraph 3 (starts with “In addition, the cost of transport...”) It would be helpful to have some explanation of how power is measured. Is this based on a calculation of energy consumption in the simulation? Based on forces and velocities from the simulation? (Etc.)
7. Results, paragraph 4 (starts with “Fig. 2 clearly shows...”) The time range for T_{bout} is small in absolute terms (0.2s), but a change from 0.2s to 0.4s represents a doubling in time. How does this play out over swimming speed? If the smaller values all are associated with the slowest swimming speed, then it would mean that T_{bout} is *not* constant. This is a place where a formal statistical analysis would support the authors' claims. For instance, if there is no change in T_{bout} with swimming speed, then a linear regression should have a non-significant slope (i.e., a hypothesis test indicates that we cannot conclude there is a relationship between swimming speed and T_{bout}). A non-significant regression slope in a linear mixed effects model, which would account for variation across individual fishes by including individual as a random effect, would provide the necessary support.
8. Results, paragraph 4 (starts with “Fig. 2 clearly shows...”) As with T_{bout} in comment #6, the authors claim that frequency is “statistically constant,” but no detail on statistical analyses has been provided. A similar linear mixed effects modeling approach is suggested.
9. Results, paragraph 4 (starts with “Fig. 2 clearly shows...”) Again, no statistical support has been provided for claims about DC or amplitude. Linear mixed effect models again would assist with the DC analysis. For amplitude, the authors may need to consult with a statistician to discuss the best approach to statistically support the change in slope indicated.
10. Figure 2 – A legend indicating the meaning of the colors and shapes would greatly facilitate reader understanding. And, what does the shading between speeds 0-1 BL/s indicate? Do the error bars represent standard error or standard deviation?
11. Figure 2 – The amplitude of the simulation does not match real fish tailbeat amplitude particularly well (overestimating amplitude between 0-1 BL/s, then decreasing and underestimating amplitude during higher swimming speeds). To be clear, I think the deviation between simulation and reality is within expectations, I do not believe this has been discussed in the manuscript – where does this difference come from? What impact may it have on the cost of transport analysis?

12. Discussion, paragraph 3 (“...it may also be explained considering that CoT minimization might not be needed at such low swimming velocities...”) I would further encourage the authors to consider how often fish swim in this mode at low speeds – if it is uncommon, then perhaps there has not been selection for minimizing energy costs. Is this the case?
13. Methods, section A – It would be helpful to have a sense of the Reynold’s number for the live fishes here. The authors later report the Re range for the simulation, but a value range for the experimental data would help readers understand how well the simulation matches the experimental conditions.
14. Methods, section B – The flume here is extremely shallow – only 3-4 times deeper than the fish’s body depths. The potential for wall and surface effects here concerns me. The authors do mention small wall effects, but it is unclear whether this is relative to the sides or bottom of the tank. And, there is no mention of whether surface effects are important. Likewise – where were the fish swimming in the water column? With a tank this shallow, the fish would need to be very centered (vertically) to avoid wall and surface effects.
15. Methods, section C – Unclear what is meant by “groups.” Were multiple fishes studied at once? In that case, did the fishes school? Which fish in the school did the data come from? Because the wake of fish at the front of a school influences flow felt by fish in the back, and because we know that fish kinematics change when entrained on vortices like those in fish wakes (see Liao 2007, Fig 1), average kinematics across a school would not reflect what an individual fish does during burst-and-coast swimming.
16. Methods, section E, paragraph 2 – What data were filtered? CoT?
17. Methods, section E, paragraph 4 – An assumption baked into this burst-and-coast assembly process is that the transition between phases is instantaneous. But in real fishes, there would be some kind of transitional flow/wake as the fish switches between bursting and coasting. My understanding is that these would be very difficult – if not completely impractical – to simulate, so I do believe that the authors have used an appropriate approximation with this assembly approach. Even so, it would be helpful to see some comments explaining the limits of the model and how these differences may influence the performance of real fishes, especially since the key conclusion drawn is that real fishes are using optimal kinematics.
18. General – it would be helpful for readers of more biological backgrounds if the authors made a distinction between mechanical costs of transport (those studied here) and metabolic costs of transport. While a fish’s gait may be optimal in the mechanical sense, that does not necessarily mean that the gait is simultaneously providing the lowest metabolic cost. I do not think this distinction lessens the importance of this work – rather, it provides more context and sets up more ways this work can fit into future investigations.

References:

Harrison XA, Donaldson L, Correa-Cano ME, Evans J, Fisher DN, Goodwin CED, Robinson BS, Hodgson DJ, and Inger R. 2018. A brief introduction to mixed effects modelling and multi-model inference in ecology. *PeerJ* 6: e4794.

Liao J. A review of fish swimming mechanics and behavior in altered flows. *Phil Trans R Soc B* 362: 1973-1993.

Zuur AF, Ieno EN, Walker N, Saveliev AA, and Smith GM. 2009. *Mixed effects models and extensions in ecology with R*. New York: Springer.

We thank the referees for their thoughtful reviews. Please find below our replies (in blue) following each question/comment.

Referee 1:

1. The main result seems to be the statement that at high speeds, burst and coast swimmers minimize the energy cost by selecting the appropriate parameters. The authors state, “This is a remarkable observation, as it shows that fish in the range of cruise swimming speeds constantly optimize their CoT.” I’m not sure I agree with the strength of this statement. Assuming the species evolved to swim with some sort of efficiency in this range, it’s hardly surprising. It sort of strikes me as an “it works because it works” type of result. Especially given that the authors state that the optimal CoT involves swimming with a consistent tail beat frequency and amplitude. It’s just as likely that biomechanics determine these parameters, and at some range of swimming speeds, they become optimal.

We understand the point raised by the referee. However, optimality is not straightforward in the multidimensional space navigated by living organisms, where locomotion is just one element of their everyday trade-offs. More specifically, we are excited by the observation and its agreement with the simulation for a twofold reason: First, unlike in continuous swimming where fish basically deal with a two-dimensional parameter space consisting of tail-beat frequency and amplitude, in burst-and-coast swimming, fish have to deal at least with a four-dimensional parameter space (shown in Fig.2: T_{bout} , DC, Frequency, Amplitude) at an arbitrary speed. The optimization of burst-and-coast swimming is thus extremely complex, especially considering that the cost of transport (CoT) can hardly be sensed directly by the fish during swimming. Second, fish have to deal with many other constraints that might not be, a priori, necessarily compatible with optimizing swimming energy. For instance, the intermittence of burst-and-coast swimming has also been invoked for a sensing reason (Olive et al. 2016). It turns out that in the case of this work, the intermittent swimming kinematics is, in a certain range of swimming regimes, exactly what optimizes swimming gaits.

We have emphasized in the revised version that before the present work we didn’t know whether fish aim to optimize the CoT during burst-and-coast swimming. Also, another non-trivial point was the question of whether fish can successfully optimize CoT in such a complex landscape of control parameters and indirect feedback. It is in this sense that we labelled as remarkable the capability of fish to optimize.

2. Section II notes that the swimming speeds for figure 1 are 1.15 and 1.65 BL/s, however, the caption of Fig. 1 states 1.15 and 1.9 BL/s. Please correct whichever is incorrect.

It's 1.9 BL/s. The error has been corrected in the revised version.

3. Introduction Page 1 paragraph 1 line 8: Misspelled 'environment'

Corrected.

4. Introduction Page 1 column 2: 'We show that, instead of modulating the frequencyand amplitude'. Define Tbout when you first mention it.

Corrected.

5. Figure 2: When talking about figure 2 in results, it would be easier to talk about them in the way they are listed. So maybe move the duty cycle to figure D or talk about it before.

We have revised the description order in this paragraph, by moving the duty cycle DC forward.

6. Methods Section C: 'Bodylength of modle fish is 2cm' (typo in "model")

Corrected.

7. There are spaces before ":" throughout the paper (i.e. "swimming kinematics : the"). Remove the space. Also, there are lots of colons in the paper. Maybe the authors should consider a more diverse rhetorical/grammatical repertoire.

We have revised the text to limit the usage of colons.

8. Discussion, paragraph 3: In the final sentence of the paragraph, remove the comma after the mdash. This happens again in the following paragraph. Perhaps this is an unconventional style of the journal, which will, of course, refer to its own style guide. I'm just noting it, as I have not encountered it before.

Corrected.

Referee 2:

1. Introduction, end of second paragraph (“Overall, there is...open question”) – “gait adaption” should be “gait adaptation.” The original is not wrong, but “adaptation” is the more accepted spelling.

We corrected this point in the new version of the manuscript.

2. Introduction, end of second paragraph (“Overall, there is...open question”) – unclear what is meant by “gait adaption in real animals is an open question.” A sentence or two expanding on what the authors mean would help reader understanding. Is the point of this paragraph that burst-and-coast swimming is often treated as an optimization problem in computational settings, but it is unclear if real animals use optimal cadences of movement?

We agree with referee 2 that this sentence might have been confusing. It has been modified in the new version. The sentence now reads: “Overall, there is a strong lack of experimental data and parametric studies concerning intermittent swimmers, making the description of speed modulation strategies for these animals an open question.”

3. Figure 1, panel B – The fish silhouettes in panel B are somewhat confusing. Specifically, the fish’s tail jumps from an extreme lateral excursion toward the bottom of the page in the 5th panel to nearly the opposite lateral extreme in the 6th panel. If this reflects real motion at some specific time interval, then a note that the tailbeat motion is unsteady, so you cannot expect the same amount of movement between panels, would greatly aid reader understanding.

There is a certain degree of irregularity even within a burst. In order to clarify and help the reading of Fig. 1, we added two movies as supplementary materials to the revised version.

4. Figure 1, panel E – While burst/coast bouts are pretty clear in C and D, it is unclear how burst-and-coast patterns are defined in this panel. For example, can we really tell that the tiny blue tick at $t = 2.5$ is a “coast”? Is it typical that the number of half-tailbeats in a bursting period are this irregular?

We thank the referee to point this out. We had missed to include the explanation on how the tail beat signal was processed. We used the derivatives of the tail position, i.e. the tip velocity and acceleration to identify the coasting phases. We have included the following sentence to the paragraph where T_{burst} , T_{coast} and T_{bout} are defined: "The duration of bursts and coasts are measured from the tail beating signals using the velocity and acceleration of the tail tip, which tend to zero during the coasting phase. We quantitatively defined the fish to be coasting when tail tip velocities and accelerations were lower than $\sim 10\%$ of their maximum values."

5. Figure 1 caption and Results paragraph 2 – how is a tailbeat being defined? A tailbeat cycle consists of a complete sine wave of amplitude change, based on the motion of the tail tip. From Figure 1C, it looks like the authors may be using “tailbeat” to refer to half a tailbeat cycle. An explicit definition would greatly clarify the authors’ meaning.

We have clarified the definition of tail beat in the revised version. In continuous swimming, one tailbeat cycle refers to the fish completing two strokes respectively on each side of the body. Here in intermittent swimming, we adopt this definition inside the burst phase. Note that as shown in figure 1C, when the fish only completes a stroke during the burst phase, it is counted as fish completes a HALF tail-beat cycle.

6. Results, paragraph 3 (starts with “In addition, the cost of transport...”) It would be helpful to have some explanation of how power is measured. Is this based on a calculation of energy consumption in the simulation? Based on forces and velocities from the simulation? (Etc.)

We have clarified the definition of power in the manuscript by adding a specific section in the Materials and Methods part.

7. Results, paragraph 4 (starts with “Fig. 2 clearly shows...”) The time range for T_{bout} is small in absolute terms (0.2s), but a change from 0.2s to 0.4s represents a doubling in time. How does this play out over swimming speed? If the smaller values all are associated with the slowest swimming speed, then it would mean that T_{bout} is not constant. This is a place where a formal statistical analysis would support the authors’ claims. For instance, if there is no change in T_{bout} with swimming speed, then a linear regression should have a non-significant slope (i.e., a hypothesis test indicates that we cannot conclude there is a relationship between swimming speed and T_{bout}). A non-significant regression slope in a linear mixed effects model, which would account for variation across individual fishes by including individual as a random effect, would provide the necessary support.

We thank the Reviewer for the suggestion to use a linear mixed effects model. We have used this approach to analyze the variation with U of four parameters: T_{bout} , F_r , \bar{A} and DC . In addition, we implemented pairwise t-tests between 3 groups: slow, medium and fast swimming speeds. A detailed description of the statistical analysis can be found in the supplementary material, a short description is included in section IV.D, and the discussion in section III is amended in view of the results of the analysis. Fig 2 has been updated with linear fits.

The variation T_{bout} is indeed the least clear. If calculated over the entire range of velocities, the fixed effect slope is significant ($p=0.0364$). However, pairwise t-tests show that only the high-speed regime has a statistically significant difference: the values of T_{bout} at high speed are

some 30% larger than at the low speed. For comparison, the amplitude \bar{A} increases by a factor greater than 2.5. Further to that, when the fixed effect slope of T_{bout} is calculated over the range of speeds between 0 and 2 BL/s, the slope is even smaller and it is insignificant ($p=0.3119$). We conclude that the full burst-and-coast cycle depends weakly on the swimming speed: fish do not modulate the time between two phases of action, unless the imposed velocity is approaching the upper limit for sustained swimming.

8. Results, paragraph 4 (starts with “Fig. 2 clearly shows...”) As with T_{bout} in comment #6, the authors claim that frequency is “statistically constant,” but no detail on statistical analyses has been provided. A similar linear mixed effects modeling approach is suggested.

The slope of F_i is statistically insignificant, and all methods that we applied agree on that. This information is included in the manuscript and the supplementary material.

9. Results, paragraph 4 (starts with “Fig. 2 clearly shows...”) Again, no statistical support has been provided for claims about DC or amplitude. Linear mixed effect models again would assist with the DC analysis. For amplitude, the authors may need to consult with a statistician to discuss the best approach to statistically support the change in slope indicated.

In addition to the regression over the entire range of velocities, we used pairwise t-tests between slow, medium and fast swimming. Only the slow swimming amplitude turned out to be significantly different. Consequently, we fitted a piecewise-linear function and found that the amplitude slope between medium and high speeds is not statistically significant. This is a sign of saturation.

10. Figure 2 – A legend indicating the meaning of the colors and shapes would greatly facilitate reader understanding. And, what does the shading between speeds 0-1 BL/s indicate? Do the error bars represent standard error or standard deviation?

A legend has been added to Fig. 2 for more clarity. Concerning the shading region, it is referred to as “the low speed region” but this definition does not appear in the text. This oversight has been corrected in the revised manuscript. The error bars represent standard deviation. This information has also been added to the text.

11. Figure 2 – The amplitude of the simulation does not match real fish tailbeat amplitude particularly well (overestimating amplitude between 0-1 BL/s, then decreasing and underestimating amplitude during higher swimming speeds). To be clear, I think the deviation between simulation and reality is within expectations, I do not believe this has been discussed in the manuscript – where does this difference come from? What impact may it have on the cost of transport analysis?

Thank you for pointing this out. We have added a discussion of the error source in amplitude prediction in the revised version. We think the amplitude error is probably because: 1) the factual fish deformation cannot be precisely reproduced by the sinusoidal function used for the numerical model; 2) When modeling the red nose tetra fish, we simplified its forked tail by a triangular one, hence the model fish tail, with slightly larger area, may generate relatively stronger lateral recoil, neutralizing the effective amplitude. Nevertheless, the prediction on the amplitude still suggests that fish don't rely on an amplification of the tail-beat amplitude to propel faster, which is similar to the trend observed in the experiment.

On the other hand, in the low-speed range, the overestimation of the amplitude may be explained by a limitation of our numerical model: as the fish performs only a single stroke during the burst phase in the low-speed range, the transition processes between the burst- and the coast- phases that are not included in the model may become no longer negligible. Here, rather than providing a specific explanation for the overestimating amplitude (the errors of the amplitude and DC are considered coupled) between 0-1 BL/s, we generally discuss that the numerical model might be unfit for low speed range prediction.

12. Discussion, paragraph 3 (“...it may also be explained considering that CoT minimization might not be needed at such low swimming velocities...”) I would further encourage the authors to consider how often fish swim in this mode at low speeds – if it is uncommon, then perhaps there has not been selection for minimizing energy costs. Is this the case?

Concerning the effect on the CoT, we agree with the referee that it is reasonable to expect that fish are most likely not optimizing CoT in this low-speed range.

13. Methods, section A – It would be helpful to have a sense of the Reynold’s number for the live fishes here. The authors later report the Re range for the simulation, but a value range for the experimental data would help readers understand how well the simulation matches the experimental conditions.

This information has been added to the new version of the manuscript.

14. Methods, section B – The flume here is extremely shallow – only 3-4 times deeper than the fish’s body depths. The potential for wall and surface effects here concerns me. The authors do mention small wall effects, but it is unclear whether this is relative to the sides or bottom of the tank. And, there is no mention of whether surface effects are important. Likewise – where were

the fish swimming in the water column? With a tank this shallow, the fish would need to be very centered (vertically) to avoid wall and surface effects.

The experimental setup is the same used in our group in previous studies. We refer the reader to our previous paper Ashraf et al. "Simple phalanx pattern leads to energy saving in cohesive fish schooling". *Proceedings of the National Academy of Sciences*, 114(36) :9599–9604 (2017), for a detailed discussion on the swimming configuration of the fish, especially the exact position of the fish in the flume and the wall effects. This reference is already cited in this section.

15. Methods, section C – Unclear what is meant by "groups." Were multiple fishes studied at once? In that case, did the fishes school? Which fish in the school did the data come from? Because the wake of fish at the front of a school influences flow felt by fish in the back, and because we know that fish kinematics change when entrained on vortices like those in fish wakes (see Liao 2007, Fig 1), average kinematics across a school would not reflect what an individual fish does during burst- and-coast swimming.

The experiments were only conducted on a single swimming fish. Groups refer to populations of the same generation and raised in the same conditions.

16. Methods, section E, paragraph 2 – What data were filtered? CoT?

We have clarified this in the revised version. Fish propel themselves in strokes, thus the time sequences of speed, power and CoT are in a periodic fluctuation pattern. In the analysis, to remove those fluctuations we smoothed the time sequences of speed, power and CoT, by a low-pass filter function.

17. Methods, section E, paragraph 4 – An assumption baked into this burst-and-coast assembly process is that the transition between phases is instantaneous. But in real fishes, there would be some kind of transitional flow/wake as the fish switches between bursting and coasting. My understanding is that these would be very difficult – if not completely impractical – to simulate, so I do believe that the authors have used an appropriate approximation with this assembly approach. Even so, it would be helpful to see some comments explaining the limits of the model and how these differences may influence the performance of real fishes, especially since the key conclusion drawn is that real fishes are using optimal kinematics.

To our knowledge, at least three transition factors may produce error. Firstly, at the same speed, the drag on a undulating fish body is much higher than that on a gliding fish body. When fish stops undulating and starts gliding, its boundary layer transits from an "undulating pattern" to a "gliding pattern", while fish stops gliding and strat undulating, the fish boundary layer transits reversely. Such drag transition is not modelled our prediction. Secondly, when fish starts the burst-phase, fish needs to activate both its body deformation and its surrounding fluids, hence the initial tail-beat may not generate thrust as effectively as succeeding tail-beats. Third, in

reality, there are fluctuations in the swimming speed of fish. Fish may utilize the fluctuation of speed by ending its burst process at peak speed, in order to initialize a coast-phase with additional momentum and further reduce CoT.

General – it would be helpful for readers of more biological backgrounds if the authors made a distinction between mechanical costs of transport (those studied here) and metabolic costs of transport. While a fish's gait may be optimal in the mechanical sense, that does not necessarily mean that the gait is simultaneously providing the lowest metabolic cost. I do not think this distinction lessens the importance of this work – rather, it provides more context and sets up more ways this work can fit into future investigations.

This is a very interesting question raised here by referee 2 and this specific problem is actually the subject of a new study we have been working on that should be submitted soon. We added a new sentence to the revised version addressing this point: “It has to be noted that the CoT as defined in the text only takes into account the mechanical cost of the swimmers. A forthcoming study will focus on the consequence of considering the additional “metabolic” cost to the conclusions raised in that work”.

REVIEWERS' COMMENTS:

Reviewer #1 (Remarks to the Author):

The authors revisions have improved the paper. The discussion in the rebuttal re: optimization was, honestly, a stronger argument than the paper itself offers. While strengthened in the revised manuscript. My only recommendation, at this point, is to make it even more clearly, if possible. However, overall, it is a fine manuscript and has been improved by the revision.

Reviewer #2 (Remarks to the Author):

Thank you to the authors for the thorough response letter and revisions. Regarding my major comments: the new statistical analyses appear sound and strengthen the authors' conclusions, and the additional information provided throughout has greatly clarified the authors' meaning. All of my concerns with the manuscript have been addressed, and I now can enthusiastically recommend the paper for publication.